# The Role of Fecal Microbiota Transplantation in IBD

**DOI:** 10.3390/microorganisms12091755

**Published:** 2024-08-23

**Authors:** Fabrizio Fanizzi, Ferdinando D’Amico, Isadora Zanotelli Bombassaro, Alessandra Zilli, Federica Furfaro, Tommaso Lorenzo Parigi, Clelia Cicerone, Gionata Fiorino, Laurent Peyrin-Biroulet, Silvio Danese, Mariangela Allocca

**Affiliations:** 1Department of Gastroenterology and Endoscopy, IRCCS Ospedale San Raffaele, Vita-Salute San Raffaele University, 20132 Milan, Italy; 2Department of Gastroenterology and Endoscopy, Santa Casa de Misericordia de Porto Alagre, Porto Alegre 90020-090, Brazil; 3Department of Gastroenterology and Digestive Endoscopy, San Camillo-Forlanini Hospital, 00152 Rome, Italy; 4Department of Gastroenterology, Nancy University Hospital, F-54500 Vandœuvre-lès-Nancy, France; 5INSERM, Nutrition-Genetics and Exposure to Environmental Risks Research Unit (NGERE), University of Lorraine, F-54000 Nancy, France; 6INFINY Institute, Nancy University Hospital, F-54500 Vandœuvre-lès-Nancy, France; 7Fédération Hospitalo-Universitaire CARE, Nancy University Hospital, F-54500 Vandœuvre-lès-Nancy, France; 8Groupe Hospitalier Privé Ambroise Paré—Hartmann, Paris IBD Center, F-92200 Neuilly sur Seine, France; 9Division of Gastroenterology and Hepatology, McGill University Health Centre, Montreal, QC H4A 3J1, Canada

**Keywords:** inflammatory bowel diseases, fecal microbiota transplantation, dysbiosis, microbiota

## Abstract

Gut microbiota dysbiosis has a critical role in the pathogenesis of inflammatory bowel diseases, prompting the exploration of novel therapeutic approaches like fecal microbiota transplantation, which involves the transfer of fecal microbiota from a healthy donor to a recipient with the aim of restoring a balanced microbial community and attenuating inflammation. Fecal microbiota transplantation may exert beneficial effects in inflammatory bowel disease through modulation of immune responses, restoration of mucosal barrier integrity, and alteration of microbial metabolites. It could alter disease course and prevent flares, although long-term durability and safety data are lacking. This review provides a summary of current evidence on fecal microbiota transplantation in inflammatory bowel disease management, focusing on its challenges, such as variability in donor selection criteria, standardization of transplant protocols, and long-term outcomes post-transplantation.

## 1. Introduction

Inflammatory bowel diseases (IBD) are chronic conditions affecting the gastrointestinal tract. They result from complex interactions between genetic predisposition, environmental factors, and immune responses [1]. A common feature of IBD is gut dysbiosis, an imbalance in the intestinal microbiota [1]. Studies in both animals and humans suggest that an altered microbiome or an abnormal immune response to the microbiome contributes to intestinal inflammation [2,3]. Patients with IBD typically have reduced microbiota diversity and lower levels of anti-inflammatory bacteria, with an increase in potentially harmful microbes. This dysbiosis perpetuates inflammation, exacerbating disease severity [4,5,6,7]. Fecal microbiota transplantation (FMT) has proven to be effective in *Clostridioides difficile* infection (CDI) [8,9]. After its successful application in CDI, FMT has emerged as a potential treatment for IBD. Unlike conventional therapies targeting the immune response, FMT modifies the microbial environment, potentially offering a safer approach [8]. Studies, including randomized controlled trials and meta-analyses, have shown that FMT can lead to clinical and endoscopic improvements in ulcerative colitis (UC) and Crohn’s disease (CD) [9]. However, heterogeneity in FMT procedures, such as timing, dosage, and delivery methods, as well as variations in donor stools, does not allow us to draw definitive conclusions regarding its efficacy [9]. This narrative review provides an overview of the available literature on the topic, focusing on FMT safety and efficacy, questioning the aspects of donor selection criteria’s variability, FMT protocols’ standardization, and long-term outcomes post-transplantation.

## 2. Dysbiosis in IBD

Intestinal microbial dysbiosis is a common feature of IBD, and it is caused by factors like genetics, inflammation, infections, and dietary habits [10]. Dysbiosis is generally marked by a decrease in beneficial microbes, an increase in potentially harmful microbes, and a general decline in microbial diversity [11,12,13]. 

The proper functioning of the gut microbiota is based on a steady ratio between the bacteria from the phyla *Firmicutes*, *Bacteroidetes*, *Acinetobacteria*, and, to a lesser degree, *Proteobacteria* [14]. Dysbiosis arises when there is a large shift in the composition ratio between these phyla or the expansion of new bacterial groups. This imbalance, characterized by reduction of microbial diversity and outgrowth of *Proteobacteria*, can be disease-promoting and can contribute to disease severity [15,16].

Inflammation and microbial dysbiosis commonly occur concurrently in patients with IBD. Thereby it remains unclear if microbial dysbiosis is an IBD’s causative factor or simply a consequence of the inflammation [17]. IBD etiopathogenesis is not fully known, and the impact of underlying gut microbiota dysfunction on the condition remains uncertain [1]. 

Nonetheless, the relationship between the gut microbiota and the host is crucial for maintaining immune function [18]. In fact, many genetic markers linked to IBD are related to the interaction between immunity and microbiota [19,20,21]. Hence, environmental changes, such as antibiotic overuse and dietary modifications, can disrupt gut microbiota composition, leading to dysbiosis and disrupting immune tolerance, potentially triggering or exacerbating CD and UC [22,23]. 

Notably, IBD disease activity is most pronounced in regions with the highest bacterial populations, such as the colon, and in areas of fecal stasis, like the terminal ileum or the rectum. Interestingly, fecal diversion is an effective treatment strategy for CD, often leading to remission in the diverted segment of the bowel [24]. Furthermore, restoring bowel continuity and re-exposing the area to the fecal stream typically results in postoperative recurrence of CD [25,26].

Data from various animal models strongly support the hypothesis that an altered microbiome or an abnormal immune response to the microbiome plays a crucial role in the development of intestinal inflammation [2,27]. Indeed, the gut microbiota composition differs between patients with IBD and healthy individuals, as well as between UC and CD patients [3]. 

There are many microbes with pathogenic qualities that proliferate within or potentially contribute to the inflammation of IBD patients. These microbes are referred to as “pathobionts” and are believed to contribute to the initiation or the exacerbation of IBD [28]. The term pathobiont, introduced in 2008, describes a gut commensal bacterium with the potential to turn pathogenic [29]. To broaden this initial definition and explain the mechanisms through which various pathobionts operate, “IBD pathobionts” are today defined as commensal microorganisms capable of causing or aggravating IBD through their pathogenic potential or niche-seeking behaviors [30].

Chronic inflammatory conditions, such as IBD, may drive the evolution of a dysbiotic microbiota, which includes pathobionts that preferentially persist in and contribute to sustaining the inflamed gut environment [10].

Pathobionts use diverse mechanisms to potentially promote IBD, and they are thought to cause disease in susceptible hosts. For example, *Bilophila wadsworthia* can cause colitis in genetically susceptible mice under a high-fat diet but does not affect healthy mice [31]. Inflammatory gut environments may give a selective advantage to certain microbes, such as *Escherichia coli*, which thrive under oxidative stress, contributing to sustained inflammation [32]. *Bacteroides fragilis* is another potential pathobiont, found abundantly in IBD patients’ biofilms and able to persist in harsh conditions [33]. 

In CD, the role of *Clostridium innocuum*, *Atopobium parvulum*, *Ruminococcus gnavus*, and *Debaryomyces hansenii* has been described [10]. *C. innocuum* can migrate from the ileal lumen to the mesenteric adipose tissue through twitching motility, causing the expansion of creeping fat around the ileum, thereby heightening the risk of fibrosis and strictures [34]. *R. gnavus* ferments sulfur-containing amino acids, leading to an excess of H_2_S that disrupts disulfide bonds in mucus, leading to inflammation [35]. *R. gnavus*, a common gut commensal with several strain variants, tends to proliferate in CD patients, leading to increased mucus degradation and inflammatory response through interactions with dendritic cells [36,37,38]. The fungus *D. hansenii* localizes to mucosal wounds and is more prevalent in the inflamed tissues of CD patients. This fungus undergoes phagocytosis by macrophages and this promotes inflammation and reduces wound healing [39].

Similarly, pathobionts may have a central role in UC. For instance, the gut commensal *Phocaeicola vulgatus* (formerly *Bacteroides vulgatus*) secretes proteases and elastases, which can increase intestinal permeability, and its presence and activity correlate with UC severity [40]. Moreover, *E. coli* strains categorized as ExPEC from the B2 phylogenetic group express genes encoding toxins (α-hemolysin) and adhesins (FimH) that disrupt intestinal epitalial cells’ tight junctions leading to dendritic cell death and colitis exacerbation [41,42]. On the other hand, *Candida albicans*, a prominent gut mycobiota member, can contribute to UC severity through the production in its hyphal phase of candidalysin, a toxin promoting intestinal inflammation [43,44]. 

Pathobionts may be difficult to detect: in order to identify microbes that may promote IBD onset and severity, isolation and identification from the gut microbiota are crucial. Culture-based methods, like selective media and whole-genome sequencing, and culture-independent methods, such as metagenomics, are used [10]. Distinguishing pathobionts from harmless microbes is challenging. Methods like mucosal washes, detecting microbe-specific antibodies, and flow cytometry help in identification. To characterize pathogenic potential, techniques include whole-genome sequencing, in vitro cell culture infection assays, and in vivo testing in genetically susceptible mouse models. These methods help study the interactions and pathogenicity of pathobionts in IBD [10].

As a result of the alterations arising in IBD patients’ intestines, when compared to the microbiota of healthy individuals, samples from patients with IBD exhibit decreased overall diversity and a lower abundance of beneficial bacterial strains. Chronic inflammation in IBD alters gut conditions, increasing oxidative stress and iron depletion while limiting carbohydrate access [45]. This leads to decreased bacterial diversity, reduced Firmicutes abundance, and increased instability over time [4]. Analyses of the gut microbiota in IBD patients reveal characteristic dysbiosis, with specific bacterial species increasing or decreasing [4]. Anti-inflammatory bacteria such as *Faecalibacterium prausnitzii*, *Clostridium groups IV* and *XIVa*, *Bacteroides*, *Sutterella*, *Phascolarctobacterium*, *Roseburia*, *Bifidobacterium* species are notably decreased [4,5,6,7]. These bacteria are linked to anti-inflammatory effects, suggesting their reduction exacerbates IBD symptoms [46]. Specifically, *F prausnitzii,* a key anti-inflammatory bacterium, is significantly reduced in IBD patients, correlating with disease activity and remission phases [47]. 

On the other hand, specific pro-inflammatory bacteria, like adherent-invasive *E. coli*, *Proteobacteria*, *Pasteurellaceae*, *Veillonellaceae*, *R. gnavus*, and *Fusobacterium*, are increased in these conditions [4].

Additionally, alterations extend to fungal and viral communities, with increases in pro-inflammatory fungi like *C. albicans* and bacteriophages and decreases in beneficial fungi like *Saccharomyces cerevisiae* [5]. *Malassezia restricta*, a common skin-resident fungus, is significantly increased in the mucosal samples of CD patients [48]. Moreover, analysis of fecal samples revealed that the gut virome of IBD patients is characterized by an expansion of *Caudovirales bacteriophages*: this virus was found to be associated with reduced bacterial diversity in UC patients, findings associated with intestinal inflammation [49,50]. Furthermore, the microbiome of patients with ileal CD is marked by an increase in fungi at the expense of bacteria, whereas patients with UC and CD without ileal involvement display reduced fungal diversity [51]. 

This dysbiosis creates a vicious cycle, perpetuating inflammation and disease severity: inflammatory gut environments may give a selective advantage to certain microbes, contributing to sustained inflammation [5] (Figure 1).

As previously mentioned, the gut microbiota composition differs between patients with IBD and healthy individuals [3]: Gevers et al. found increased *Pasteurellaceae*, *Veillonellaceae*, *Neisseriaceae*, *Fusobacteriaceae*, and *E. coli*, and decreased *Bacteroides*, *Clostridium nexile*, *Clostridium bolteae*, *Faecalibacterium*, *Roseburia*, *Blautia*, *Ruminococcus*, and *Lachnospiraceae* in newly diagnosed pediatric CD patients. This suggests that microbiota alterations may precede clinical disease and occur independently of treatment. CD is associated with a more unstable gut microbiota compared to UC, with significant loss of butyrate-producing organisms like *Faecalibacterium* and *Christensenellaceae* [52].

The studies on microbiome in IBD may also be relevant from the perspective of therapy: Yilmaz et al. demonstrated that changes in microbiota composition correlated with treatment response in CD and relapse risk post-surgery [53]. Responsiveness to TNFi therapy was associated with *Bifidobacterium*, *Collinsella*, *Lachnospira*, *Lachnospiraceae*, *Roseburia*, and *Eggerthella*. Post-surgical CD patients with inactive disease showed reductions in *Parabacteroides* and *Clostridiales* and increases in *Enterobacteriaceae* [53].

Multi-omics analyses have shown distinct differences in gut microbiota and their metabolites between active and remission phases of CD and UC. These findings highlight the potential for developing personalized treatment plans based on gut microbiota profiles and the importance of targeting host-microbe interactions for novel IBD therapies [54].

## 3. Fecal Microbiota Transplantation

FMT involves transferring minimally manipulated, pre-screened donor stool into the gastrointestinal tract of a patient. This procedure aims to ameliorate dysbiosis by increasing overall microbial diversity and restoring microbiota functionality [55].

FMT re-establishes balanced intestinal flora by introducing fecal matter from healthy donors into a compromised gastrointestinal tract. This intervention treats various conditions by markedly enhancing bacterial diversity, thereby cultivating a healthier microbiota profile akin to that of healthy donors, with sustained benefits. FMT has gained increasing interest in recent years due to evolving methodologies and expanding clinical indications [56].

Since 2013, several studies have supported FMT safety and high efficacy in preventing the recurrence of CDI [57,58,59]. Antibiotic use often leads to recurrent CDI by disrupting gut microbiota. FMT restores the gut microbiota to its pre-antibiotic state, and it has proven to be more effective than standard antibiotics like vancomycin and fidaxomicin. Indeed, FMT is now recommended in the CDI treatment international guidelines for patients with multiple recurrences of CDI who have failed appropriate antibiotic treatments [60].

Following its success in CDI management [55], FMT has also been investigated in patients with IBD [61]. Numerous trials are ongoing worldwide to explore additional therapeutic applications (Figure 2).

Key considerations for FMT are encapsulated in the 5D framework proposed by Allegretti et al. in 2018 for the procedure’s application in CDI that may be applied in other FMT executions. This framework includes five steps: decision, donor, discussion, delivery, and discharge [55].

Patient selection is crucial, and FMT indications coming from scientific evidence must be followed. Donor-related factors and recipient-related factors need to be analyzed to achieve donor-recipient microbiota compatibility. For donors, it is important to identify individuals with high microbiota richness and a high abundance of beneficial strains [62]. Available evidence indicates that the features and the composition of donor microbiome can impact FMT’s clinical outcomes [62]: for instance, a high relative abundance of many donor taxa like *Lachnospiraceae*, *Ruminococcus*, *Akkermansia muciniphila*, and *Bacteroides* has been associated with induction of remission for FMT in UC [63,64,65]. However, current data also outline that individual taxa are not reliable predictors of clinical outcomes and that the role of the microbiome in disease progression is much more intricate [66].

The donor material can be patient-directed or obtained from universal stool banks. Universal banking, often integrated within hospital systems, allows for rigorous screening and pre-banking of donor material, minimizing delays. Both fresh and frozen donor materials have been used in clinical practice and in research settings. Recipients, on the other hand, should be assessed based on their underlying disease, genetic factors, immune status, and microbiota composition, assessing the patients’ pre-conditioning [55,62].

A rigorous donor screening must be carried out to avoid rare but serious infections. However, there is a lack of information regarding donor selection strategy: advances in technology may guide us in the donor and recipients’ selection process [62].

Discussion through informed consent must cover the risks, benefits, and alternatives to FMT. Common adverse events, like moderate fever and mild gastrointestinal symptoms, should be explained to patients [55].

FMT delivery varies depending on the clinical context, including upper endoscopy, nasoenteric tubes or capsules for upper gastrointestinal (GI) tract administration, colonoscopy, flexible sigmoidoscopy, or enemas for lower GI tract administration. Each method has specific advantages and disadvantages (Table 1).

Procedural factors are extremely relevant in the FMT process, its timing as well as its dosage, and the frequency of administration [55,62].

Finally, post-FMT care involves educating patients on infection control and antibiotic stewardship. Follow-up is essential to monitor for adverse events and recurrence, typically up to eight weeks post-transplant [55].

## 4. Fecal Microbiota Transplantation in IBD

Unlike most current therapeutic strategies that directly target the immune response, which is associated with high costs and potential adverse effects, FMT represents an alternative in the treatment of IBD by modifying the microbial environment [68]. This modification can indirectly influence the host’s immune system in a potentially safer manner [8,68]. By improving the balance of gut microbiota, FMT has demonstrated multiple benefits, leading to endoscopic and clinical improvements in UC and CD patients compared to control groups [8].

### 4.1. FMT in UC

In the last decade, several studies have been carried out showing that FMT appears to have a benefit in the induction of remission for mild to moderate UC, with improvements in endoscopic and clinical symptoms. Moreover, these studies have suggested FMT’s potential in preventing UC flares or complications [9].

Two RCTs from 2015 outlined FMT efficacy in patients with active UC. Moayyedi et al. conducted a study to evaluate the efficacy of FMT in patients with active UC. Participants were randomly assigned to receive either FMT (38 patients received 50 mL via enema from healthy anonymous donors) or placebo (37 patients received 50 mL water enema) once a week for 6 weeks. Concomitant therapies, including mesalamine, glucocorticoids, immunosuppressants, or tumor necrosis factor inhibitors, were allowed if they had been administered at a stable dose for at least 12 weeks (4 weeks for glucocorticoids). The primary endpoint was clinical remission, defined as a Mayo score of ≤2 with an endoscopic Mayo score of 0, assessed at week 7. FMT induced remission in a significantly greater percentage of patients with active UC compared to the placebo group (9 patients (24%) vs. 2 patients (5%); 95% confidence interval, 2–33%). Greater microbial diversity was observed in stool samples from FMT recipients compared to placebo (*p* = 0.02). Notably, the efficacy of FMT was influenced by the fecal donor and the duration of UC [63].

Haifer et al. evaluated, instead, the efficacy of oral lyophilized FMT in active UC patients aged 18–75. Following a 2-week antibiotic regimen, patients were assigned to either FMT or placebo capsules for 8 weeks. At week 8, 53% of patients in the FMT group achieved corticosteroid-free clinical remission accompanied by an endoscopic response (defined as a total Mayo score of ≤2, with all subscores ≤ 1 and a reduction of ≥1 point in the endoscopic subscore), compared to 15% of patients in the placebo group (difference of 38.3%, *p* = 0.027; odds ratio 5.0). In the maintenance phase, 100% of patients continuing FMT maintained remission at week 56, while none of the withdrawal group did [69].

Costello et al. performed a systematic review and meta-analysis to determine FMT efficacy and safety for inducing remission in active UC. It included 14 cohort studies and 4 RCTs [70]. The 14 cohort studies involved a total of 168 patients with active UC, ranging from mild to severe, with follow-up periods varying from 1 to 72 months. Patients generally continued their regular IBD medications during FMT, except in one study where all medications other than 5-aminosalicylates were interrupted [70,71].

Of the 168 patients, 93 (55%; 95% CI: 36.7–71.7%) achieved a clinical response, amongst whom 39 (24%; 95% CI: 11–40%) attained clinical remission, though these outcomes were defined variably across the studies. Endoscopic remission was assessed in only seven out of the 14 studies and was achieved in 16 of 56 patients. Additionally, five out of the 14 cohort studies reported antibiotics pretreatment: in these studies, treatment success was numerically higher [71,72,73,74,75]: 39 out of 58 patients (67%) experienced a clinical response, and 19 out of 58 (32%) achieved clinical remission [70].

The 4 RCTs in the Costello meta-analysis, involving a total of 277 patients, proved an achievement of clinical remission equal to 28% of patients in the donor FMT groups versus the 9% in the placebo groups (odds ratio 3.67, 95% CI: 1.82–7.39, *p* < 0.01). Clinical response was achieved in 49% of donor FMT patients compared to 28% of placebo patients (odds ratio 2.48, 95% CI: 1.18–5.21, *p* = 0.02) [70]. The studies analyzed showed heterogeneity in design with different routes of FMT administration, inclusion criteria, and follow-up periods. However, despite protocol variations, FMT appeared effective for inducing remission in UC [70].

The effectiveness of FMT in IBD was further confirmed by a systematic review and meta-analysis conducted by Paramsothy et al., which included 53 studies involving patients with UC, CD, and pouchitis. The primary outcome of clinical remission was achieved in 36% of UC patients, 50.5% of CD patients, and 21.5% of pouchitis patients. Specifically, 36% (201/555) of UC patients achieved clinical remission during follow-up. Among the 24 cohort studies included in the meta-analysis, comprising 307 individuals, the pooled proportion of UC patients who achieved clinical remission was 33% [95% CI: 23–43%], with a moderate risk of heterogeneity [Cochran’s Q, *p* = 0.001; I2 = 54%] and no publication bias. The pooled proportion of patients achieving a clinical response was 52% [95% CI: 40–64%], according to a meta-analysis involving 234 individuals from 20 cohort studies, also showing moderate heterogeneity [Cochran’s Q, *p* = 0.001; I2 = 58%] and no publication bias [76]. Meta-analysis of 4 RCTs of FMT in UC, involving 140 FMT-treated individuals, demonstrated that FMT was significantly associated with clinical remission [P-OR = 2.89, 95% CI: 1.36–6.13, *p* = 0.006], with moderate heterogeneity [Cochran’s Q, *p* = 0.188; I2 = 37%] and no publication bias. A significant association was also found between FMT and clinical response in UC patients [P-OR = 2.48, 95% CI: 1.18–5.21, *p* = 0.016], again with moderate heterogeneity [Cochran’s Q, *p* = 0.102; I2 = 52%] and no publication bias [76].

Furthermore, sub-analyses of this work evaluated the role of the number of FMT administrations: the pooled proportion of UC patients who received more than 10 infusions and achieved clinical remission was 49% [95% CI: 21–77%; Cochran’s Q, *p* = 0.246; I2 = 29%], while the remission rate observed in those who received 10 or fewer FMT infusions was equal to 27% [95% CI: 17–40%; Cochran’s Q, *p* = 0.001; I2 = 58%] [76].

While FMT appears effective for UC, its long-term durability remains uncertain, necessitating further controlled studies, especially for CD and pouchitis [76].

Based on this evidence, there is a possibility for FMT to become a therapeutic option for UC patients. Nonetheless, these data are insufficient to make us recommend FMT as a treatment for UC in routine clinical practice. Therefore, its use is now limited to the research setting [9].

Current evidence suggests that FMT benefits are temporary since after reaching positive outcomes in the short-term, disease relapses are likely: this indicates the necessity for maintenance therapy to achieve long-term efficacy. Thereby, further research is required to explore the potential of FMT as a maintenance therapy and to determine the optimal number of infusions and dosing for both induction and maintenance phases. Presently, there is no consensus on the minimum number of FMT administrations required for success [77]. Additionally, FMT efficacy in UC appears to be crucially connected to donor-recipient engraftment. Recipient markers, alongside donor markers, contribute to FMT success: it is important to investigate their role in interacting with donor fecal material and antigens [9,63] (Table 2).

### 4.2. FMT in CD

The use of FMT for CD is primarily supported by case reports and pilot studies, with a lack of large RCTs. Consequently, there is insufficient evidence to support FMT application as a treatment for CD in clinical practice [9].

One of the first pilot studies on this topic, conducted by Vermeire et al., observed no differences at week 8 post-FMT among six patients affected by refractory CD [78]. Additionally, Sokol et al. investigated the impact of FMT on sustaining remission in CD and found a non-significant reduction in the incidence of flares in the FMT group compared to the sham group. This study was the first randomized trial assessing FMT’s capacity to maintain stable disease remission in CD patients who achieved remission with corticosteroids. The FMT group exhibited a numerically greater rate of steroid-free clinical remission alongside a lower endoscopic severity index and reduced C-reactive protein levels (CRP) levels, although these differences were not statistically significant [79]. The rates of steroid-free clinical remission were 44.4% (4/9) at 10 weeks and 33.3% (3/9) at 24 weeks in the sham transplantation group, compared to 87.5% (7/8) at 10 weeks and 50.0% (4/8) at 24 weeks in the FMT group. Six weeks after FMT, the Crohn’s Disease Endoscopic Index of Severity significantly decreased (*p* = 0.03), whereas no significant change was observed after sham transplantation (*p* = 0.8). In contrast, CRP levels increased significantly 6 weeks after sham transplantation (*p* = 0.008) but remained unchanged following FMT (*p* = 0.5) [79].

A systematic review with the meta-analysis by Caldeira et al. analyzed 60 studies, with 36 included in the quantitative synthesis in patients of any age or gender diagnosed with CD, UC, and pouchitis, with or without a treatment comparator and reporting response, remission, or adverse events. Pairwise meta-analyses of six controlled trials demonstrated significant advantages for FMT over placebo, with an RR for clinical remission of 1.70 (95% CI: 1.12 to 2.56) and for clinical response of 1.68 (95% CI: 1.04 to 2.72). The overall clinical remission rate was 37%, and the clinical response rate was 54%, with adverse events reported in 29% of cases. In the study, CD patients gained more benefits than UC from the procedure. Moreover, the analysis suggested that frozen fecal material from universal donors may enhance clinical remission rates, particularly in CD patients [80].

Frozen FMT offers several advantages, including the immediate availability of fecal material, cost savings due to fewer donor screenings, and reduced structural requirements for the practice setting where the transplantation is performed [81]. Additionally, it is important to note that the freezing and thawing processes do not significantly alter the composition of the viable microbiota [82].

On the other hand, a study by Fang et al. revealed no significant difference in clinical remission trend between fresh and frozen FMT for CD patients, while a trend towards a higher clinical remission rate with frozen stool FMT compared to fresh stool FMT for UC was observed [83].

Further evidence on the topic is brought by the studies by Cheng and Tang that highlighted fresh stool FMT’s benefits [84,85]. In Chang et al.’s sub-analyses, clinical remission rates were higher with fresh stool FMT compared to frozen stool FMT (73% vs. 43%; *p* < 0.05) [84]. Furthermore, in Tan et al. study fresh fecal FMT had a higher clinical remission rate (40.9%) compared to frozen fecal FMT (32.2%) [85].

The systematic review and meta-analysis by Cheng et al. including 12 trials, proved FMT effective and safe for CD. A pooled analysis revealed that 62% (95% CI: 48–81%) of CD patients achieved clinical remission, and 79% (95% CI: 71–89%) achieved a clinical response following FMT [84].

FMT’s success may be attributed to its ability to increase the overall diversity of the enteric microbiome [84,85]. Patients prior to FMT exhibit decreased species diversity and significant microbiome compositional differences compared to their donors. However, after the procedure, clinical responders to FMT, in contrast to non-responders, develop significantly higher species diversity, more closely resembling donor microbiome configurations [84].

Further research is needed to optimize the induction and maintenance of remission in these patient populations: large RCTs are essential to establish FMT as a viable treatment approach for CD patients. In this context, the role of donor-recipient engraftment remains one of the most important aspects to analyze in terms of FMT efficacy for CD patients [86] (Table 3).

### 4.3. FMT in IBD Pediatric Population

Most studies on FMT in IBD have primarily concentrated on the adult population, resulting in less clarity regarding its efficacy and safety in the pediatric population. However, comprehending the role of FMT in pediatric IBD is crucial [87]. Microbial-based therapies could be particularly effective in treating children [83].

The possibility of using FMT as a treatment tool for IBD has been investigated in pediatric patients: the systematic review by Hsu et al. evaluated FMT in 352 pediatric IBD patients through 11 studies. One month after FMT, clinical response (defined as a reduction of 20 or more points in the Pediatric Ulcerative Colitis Activity Index (PUCAI) for UC patients or a decrease of more than 12.5-point in the Pediatric Crohn’s Disease Activity Index (PCDAI) for CD patients) was achieved in 58.8%, clinical remission (defined as PUCAI or PCDAI under 10) in 64.7% and both in 44.1% of patients. These findings suggest FMT can be effective in pediatric IBD, potentially with improved outcomes compared to adults [88].

Fang et al.’s meta-analysis included 23 cohort studies (15 in UC, 4 in CD, and 4 in both UC and CD) involving a total of 319 subjects (225 UC and 94 CD), both adults and pediatric patients, which were all treated with FMT. Subjects were administered with fresh donor stool or frozen stool, either via the upper GI, the lower GI, or both. The stool donors were healthy adults or children, family members, close friends, or volunteers [83].

A total of 67 pediatric IBD patients (47 with UC and 20 with CD) and 252 adult IBD patients (178 with UC and 74 with CD) were included in the subgroup analysis. Clinical remission was achieved by 6 of the 47 pediatric UC patients and 45 of the 178 adult UC patients. Among pediatric CD patients, 9 out of 20 achieved clinical remission, while 31 out of 74 adult CD patients did the same. The pooled estimate of clinical remission following FMT was 10% (95% CI: 0–43%) for pediatric UC and 45% (95% CI: 24–66%) for pediatric CD, compared to 26% (95% CI: 10–48%) for adult UC and 22% (95% CI: 3–52%) for adult CD. The efficacy of FMT in IBD patients was not influenced by the type of stool (fresh or frozen), delivery route, or antibiotic pretreatment. It is important to underline that data on the effects of FMT in pediatric IBD were limited, and no randomized controlled trial was included [83].

The pediatric population, with its dynamic and developing gut microbiome, can experience a more aggressive course of CD and UC compared to adults. This suggests that pediatric IBD may have a pathophysiology distinct from adult-onset IBD. Unlike adults, children have a less stable microbiome, and there is a paucity of literature on the topic [89]. Recent evidence suggests that the microbiome composition during childhood and adolescence is delicate and dynamic, and its composition is especially susceptible to environmental factors [87].

Therefore, the role of FMT in the management of pediatric IBD needs to be thoroughly assessed, as current data on this subject are extremely constrained. Available results are limited by the lack of established protocols and long-term follow-up data. Further research is necessary to elucidate the potential of FMT as a therapeutic strategy in the pediatric population (Table 4).

### 4.4. FMT for CDI in IBD

Recurrent CDI is characterized by the disruption of healthy gut microbiota, and FMT aids in restoring gut flora diversity, thereby reducing the risk of additional CDI episodes. The risk of CDI is elevated in individuals with IBD because of the underlying dysbiosis [90].

For both mild and severe cases of recurrent or refractory CDI in IBD patients, FMT is recommended as a therapeutic option. Indeed, FMT is effective for the treatment of recurrent CDI in patients without IBD as well as IBD patients [9].

In a systematic review and meta-analysis by Tariq et al., FMT was proved to be effective and safe as a therapy for CDI in IBD patients. Out of 457 adult patients, 363 experienced CDI resolution following the first FMT, resulting in a pooled cure rate of 78% (95% CI: 73–83%; I2 = 39%). When considering both single and multiple FMTs, the overall pooled cure rate was 88% (95% CI: 81–94%; I2 = 73%). However, approximately 26.8% (95% CI: 22.5–31.6%; I2 = 9%) of adult patients experienced a flare-up of IBD following FMT, and 7.3% (95% CI: 4.7–10.5%; I2 = 56%) necessitated colectomy. In pediatric patients, among 141 participants, 106 achieved CDI resolution after the first FMT, achieving a pooled cure rate of 78% (95% CI: 58–93%; I2 = 59%). The pooled cure rate for both single and multiple FMTs was 77% (95% CI: 50–96%; I2 = 63%). About 10.8% (95% CI: 5.7–18.5%; I2 = 43%) of pediatric patients experienced an IBD flare following FMT, with 10.3% (95% CI: 2.1–30.2%; I2 = 23%) needing a colectomy. The effectiveness of FMT for CDI seems to be less effective in patients with IBD patients compared to those without IBD, likely due to an already disrupted gut microbiome. Overall, FMT is very effective in preventing recurrent CDI among IBD patients, with multiple FMTs providing greater benefit for those who do not respond to a single FMT [91].

However, studies in patients with UC indicate that single-dose FMT does not effectively prevent UC flares. Hence, there is insufficient evidence to support FMT as a treatment for CDI initial episodes in IBD [9].

### 4.5. FMT Safety in IBD

Due to the lack of long-term data, the safety profile of FMT remains uncertain. Although FMT has demonstrated positive outcomes in patients with IBD in small case series and retrospective studies, it is not without risks. Despite perceptions of FMT as a “natural” treatment by both patients and physicians, potential adverse effects exist, including the risk of disease transmission from donor feces [92].

Adverse events appear to be less frequent when FMT is administered to the lower gastrointestinal tract, meaning directly via endoscopy into the terminal ileum, cecum, or sigmoid or using a rectal enema. Enteric pathogen transmission, a significant concern, is rare due to rigorous donor screening processes. However, several adverse effects have been reported in IBD patients undergoing FMT, including moderate fever and mild gastrointestinal issues. These adverse reactions can be categorized into short-term and long-term effects [93]. Short-term adverse effects include abdominal tenderness, pain, bloating, flatulence, diarrhea or constipation, borborygmus, nausea, vomiting (especially in patients receiving the oral FMT route), and transient fever. Potential long-term adverse effects are more severe and may include infections, sepsis, the transmission of enteric pathogens, or unrecognized infectious agents such as HIV or hepatitis C, which could manifest years later. Furthermore, there is limited information available on the immunologic effects of FMT, including the potential for latent infections. Additionally, FMT has been described as potentially connected to various diseases or conditions via alterations in the intestinal microbiota, with colon cancer, obesity, atherosclerosis, diabetes, asthma, non-alcoholic fatty liver disease, and autism being of particular concern [8,94].

#### 4.5.1. FMT Safety in UC

The available data on FMT in UC patients indicate that FMT carries a low risk in terms of safety in the induction of remission in mild to moderate UC. In the meta-analysis by Paramsothy et al. after FMT in UC, common adverse events were transient minor gastrointestinal symptoms, such as bloating, diarrhea and flatulence, and transient fever. Most adverse events resolved spontaneously within days after the procedure [76].

Costello et al. outlined how FMT was well tolerated in UC patients, with the most common reported adverse events being self-limiting gastrointestinal complaints. The most frequently reported serious adverse event included worsening colitis (observed in 3 out of 140 patients in the donor groups and 4 out of 137 patients in the placebo groups), three cases of small bowel CD and two cases of *C. difficile colitis*, one of which required colectomy [70]. Two of the included studies reported adverse events more extensively, with a percentage of patients in the donor arms having a minimum of one adverse event, equal to 79% versus 75% in the placebo one [95,96]. However, there were no significant differences in either the frequency or the types of adverse events between the donor and placebo groups in any of the studies [70,95,96].

#### 4.5.2. FMT Safety in CD

There are insufficient data on FMT safety in CD, and no conclusion can be drawn on the topic, given the lack of large RCTs and long-term follow-up data [9].

The Caldeira et al. meta-analysis, including 16 studies and four RCTs, reported the frequency of any adverse event equal to 26.9% [95% CI 16.5–40.6] and 48.2% [95% CI 15.4–82.6], respectively. The overall incidence of adverse events was 29.2% [95% CI 18.8–42.5]. Most of these events were mild, including symptoms such as diarrhea, abdominal pain, nausea, flatulence, and fever, which typically resolved within 24 h post-transplant [80]. In the subgroup analysis, no significant difference was found in adverse events based on the type of donor. On the other hand, a notable difference emerged when comparing the type of stool used, with a higher incidence of adverse events observed in procedures using frozen fecal material compared to fresh stool or capsules. Similarly, in the subgroup analysis by IBD type, a very low incidence of adverse events was noted in CD patients. However, these findings should be interpreted with caution, as many of the included studies did not standardize the reporting of adverse events, potentially leading to bias. Consequently, it is difficult to definitively conclude whether stool characteristics or IBD type influence the occurrence of procedure-related adverse events [80].

The systematic review and meta-analysis by Cheng et al. also assessed FMT safety in CD. The majority of adverse events were mild and self-limiting, with no major FMT-related complications reported. However, 13 serious adverse events were noted in the study conducted by Sokol et al. [79], among which there were nine cases of CD flares (six occurring in the sham group and three in the FMT group): although the incidence of flares was lower in the FMT group compared to the sham group, the difference was not statistically significant [84].

It is important to note that the small sample size and the absence of a control arm in most of the available studies hinder the ability to standardize the assessment of adverse events. Moving forward, it is essential to include more patients with long-term follow-up to thoroughly evaluate the safety of FMT treatment for CD [80,84].

Safety concerns are certainly questioned in the pediatric population as well. The systematic review by Hsu et al. assessing FMT in pediatric IBD patients showed a pooled rate of adverse events of 29% (95% CI: 15–44%), with serious adverse events at 10% (95% CI: 6–14%). Most of the events were mild, including abdominal pain, bloating, nausea, vomiting, and diarrhea, suggesting a good FMT safety profile [88].

Indeed, many of the symptoms reported as common adverse events in FMT studies can be seen in IBD patients at baseline, and multiple papers have acknowledged that it can be difficult to discern whether the symptoms are truly FMT-related or are justified by the underlying disease process [88].

In summary, while FMT shows promise as a treatment for IBD, its safety profile necessitates further investigation, particularly regarding long-term risks and adverse effects, especially serious ones, including disease exacerbation [9].

### 4.6. FMT Perspectives in IBD Management

Many therapeutic options are currently available for treating IBD, and significant progress has been made in managing these conditions. However, new and combined strategies are needed, especially for patients who fail to respond to existing treatments. In this context, further research is necessary to explore the combination of FMT with current IBD therapies, including corticosteroids, biological agents, and small molecules. This combined approach, targeting both the immune response and the gut microbiome composition, may achieve higher remission rates than individual treatments. Studies are required to assess the role of FMT as a stand-alone treatment for IBD, as well as in combination with currently available treatment approaches. Additionally, studies should investigate FMT’s potential to prevent disease onset and postoperative recurrence [9].

### 4.7. Associate Role of Nutrition and Diet

The effectiveness of gut microbiome modulation via FMT may be enhanced through additional approaches. These include implementing a supportive anti-inflammatory dietary regimen for both donors and recipients, optimizing bowel preparation and administering antibiotics before FMT, and incorporating probiotic, prebiotic, synbiotic, and postbiotic supplementation [9].

Diet may have a relevant role in enhancing the FMT effect. For instance, an RCT by Kedia et al. proved that combining FMT with an anti-inflammatory diet (FMT-AID) may be effective in inducing both clinical and endoscopic remission in patients with mild to moderate UC. AID’s main principles were the avoidance of gluten-based grains, dairy products, refined sugars, processed and red meat, and food additives while increasing the intake of fermented foods, fresh fruits, polyphenols, and vegetables, including the AhR (aryl hydrocarbon receptor) ligand-rich ones [97].

In the study conducted by Kedia, FMT-AID demonstrated greater efficacy compared to standard medical therapy in achieving clinical response (23/35, or 65.7%, vs. 11/31, or 35.5%, *p* = 0.01, OR 3.5 [95% CI 1.3 to 9.6]), remission (21/35, or 60%, vs. 10/31, or 32.3%, *p* = 0.02, OR 3.2 [95% CI 1.1 to 8.7]), and deep remission (12/33, or 36.4%, vs. 2/23, or 8.7%, *p* = 0.03, OR 6.0 [95% CI 1.2 to 30.2]) at 8 weeks. Further, the continuation of an anti-inflammatory diet alone was effective in maintaining clinical and endoscopic remission compared to standard medical therapy until 48 weeks of evaluation (6/24 (25%) vs. 0/27, *p* = 0.007) [97].

The role of diet in IBD management and the interplay between diet, microbiota, and IBD is further proved by the example of Crohn’s Disease Exclusion Diet (CDED). CDED is a restrictive diet that eliminates or reduces the intake of specific dietary components, including animal fats, gluten, dairy products, simple sugar, red and processed meats, and various food additives such as emulsifiers, sulfites, artificial sweeteners, taurine, and carrageenan. This diet is often used in conjunction with partial enteral nutrition (PEN) to help achieve disease remission by avoiding these particular dietary elements. A systematic review by Correia et al. examined the available evidence on the effectiveness of combining CDED with PEN for inducing remission in both pediatric and adult patients with active CD [98]. The use of this dietary approach implies a cause-and-effect relationship, suggesting that the components eliminated by the CDED may contribute to inflammation through disruption of the intestinal microbiota, leading to dysbiosis. By removing these elements, the diet appears to have a positive impact on the health of the intestinal mucosa [99,100,101]. Thereby, diet may have a relevant role in shaping microbiota’s composition in order to reduce or prevent the inflammatory processes caused by its alteration, potentially contributing to CD management and mucosal healing achievement, which is a critical goal in CD remission and management [100,102]. Studies are needed to provide evidence-based data on the use of diet as a complementary approach to enhance FMT effectiveness.

## 5. Discussion

The recognition of the gut microbiota as a critical factor in the development of various chronic inflammatory diseases has spurred a growing interest in the use of FMT for a wider array of conditions. Despite this, the success of FMT in these new applications has not matched the clear efficacy observed in treating CDI, which remains the sole approved use of FMT [86]. CDI has a relatively straightforward pathogenesis driven by the overgrowth of *C. difficile* due to disruption of the normal gut microbiota. FMT is effective in CDI because it reestablishes a balanced microbial community, which can outcompete and inhibit *C. difficile*, leading to the resolution of the infection [8,9]. On the other hand, IBD is characterized by a complex interplay of genetic, environmental, immune, and microbial factors, which involves complex interactions among various bacterial strains; therefore, the altered microbiota composition in IBD patients may not be fully corrected by FMT. The multifaceted nature of IBD means that FMT may not be as successful in managing IBD due to the chronic and diverse nature of its underlying causes, with some studies showing initial improvement but less consistent long-term benefits [76].

Immune system interactions are also involved with a significant role: IBDs are charactherized by chronic inflammation and immune system dysregulation [1,4]. The altered immune responses in IBD patients can limit the ability of transplanted microbiota to exert beneficial effects [1]. This contrasts with the rationale for using FMT to treat CDI, where the primary concern is microbial imbalance rather than immune dysfunction. Furthermore, IBD is often associated with mucosal barrier dysfunction [1]. Persistent inflammation and damage to the mucosal barrier can reduce the effectiveness of FMT to restore normal gut function and microbial stability. Even more, FMT protocols for CDI are relatively standardized, which contributes to their high success rate. Conversely, protocols for IBD are less established and vary between studies [70,76,80,83,84,85,88], affecting the results and leading to uncertainty regarding the best methods and modalities to carry out FMT [2].

Up-to-date studies on FMT outlined its potentiality in autoimmune diseases, behavioral diseases, metabolic disorders, and organic diseases [103,104]. Among the possible applications for its practice, IBD and, in particular, UC appears to benefit from FMT as a potentially effective treatment in both adults and children. Evidence from systematic reviews and meta-analyses underscores the potential of FMT as a viable treatment for IBD, especially for inducing clinical remission and response [9].

However, available research on this topic is composed of heterogeneous works; therefore, its use is today confined to the research setting [9]. Further research is, indeed, needed before FMT can be recognized as a standard strategy for IBD treatment, aiming to improve knowledge about short- and long-term success rates. Standardized procedures will be essential to fully realize FMT’s therapeutic potential for IBD. Future studies should focus on standardizing FMT procedures and exploring its role as maintenance therapy. However, standardizing FMT is extremely challenging, given that 20–50% of shotgun metagenomics reads cannot be mapped, 30–50% of identified genes are of unknown function, and 80–90% of metabolites cannot be identified. Furthermore, elements in the microbiota, such as viruses, archaea, fungi, and protists, are not analyzed or considered in most of the published FMT studies [105,106].

Many aspects need to be addressed and studied to truly include FMT among the possible therapeutic strategies in the future IBD management: donor-related factors, recipient-related factors, donor-recipient engraftment, FMT dose, its preparation as well as its delivery route, the frequency of its administration and the possible markers to assess its efficacy and its role in disease course [62].

The success of FMT depends on the compatibility between donor and patient microbiota, making donor screening essential. Following the CDI protocol for screening not only enhances efficacy but also helps prevent adverse events [62]. However, the risk of associated complications is still uncertain due to insufficient long-term efficacy and safety data and the lack of information regarding patient phenotype and donor selection strategy [9,62].

Achieving donor-recipient microbiota compatibility requires an assessment of both donor-related factors and recipient-related factors: both donor and recipients should be evaluated based on underlying disease, genetic factors, immune state, and microbiota composition [62]. The importance of these evaluations is further proved by findings coming from trials about FMT applications in other conditions: studies in patients with multiple sclerosis, non-alcoholic steatohepatitis, and irritable bowel syndrome suggest that FMT requires successful engraftment of donor microbiota for long-term symptom relief. Despite efforts to monitor strain engraftment levels, much remains to be understood about donor-recipient compatibility [107,108,109].

Modern technology may guide us in seeking compatibility: advances in technology, such as analytic modeling and artificial intelligence, could facilitate the selection process in finding donors and their corresponding recipients, and vice versa [62]. Specific matching models, based on analytic hierarchy processes or machine-learning approaches, have been defined as capable of preliminarily identifying the most suitable donor for a specific recipient. These findings support the advancement of efforts in this field [62,110,111].

Additionally, procedure-related factors must be addressed: available studies on FMT are extremely heterogeneous, especially from the perspective of FMT timing, administration route, dose, repetition, antibiotic pre-medication, and recipient pre-conditioning. Thereby, the modalities trough which we should provide FMT treatment remain uncertain and not standardized [62].

Furthermore, in order to evaluate FMT results, outcomes evaluation is paramount. Apart from the outcomes of clinical and endoscopic remission, microbial markers, like increased microbiome diversity, should be assessed and correlated with FMT success or failure in future studies. However, current evidence suggests that single taxa are not consistent predictors of clinical outcomes and that the role of the microbiome in the natural course of diseases is by far more complex [66]. Thereby, markers capable of predicting patient response should be found to provide and enable personalized and precision medicine approaches [9].

## 6. Conclusions

In conclusion, FMT represents a potential treatment for IBD patients, serving as a strategy to enhance clinical outcomes, including achieving endoscopic remission. Further work is required to assess the role of FMT either as a stand-alone therapy or in combination with currently available treatment approaches. Further studies to develop protocols for conducting FMT and to identify markers capable of assessing patients’ therapeutic responses are needed. IBD management requires new and combined strategies to assure patients with personalized and precise solutions to achieve disease remission. FMT has the potential to become one of these strategies.

## Figures and Tables

**Figure 1 microorganisms-12-01755-f001:**
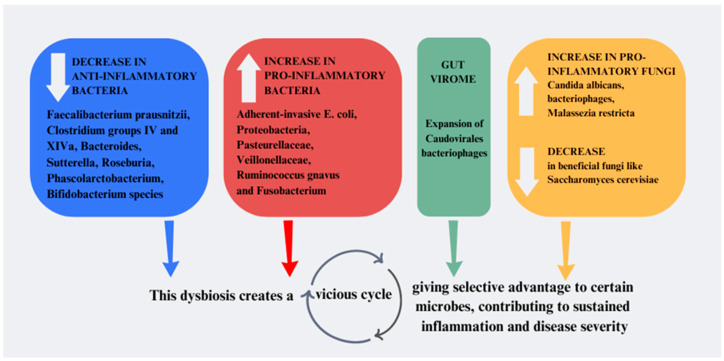
Dysbiosis in IBD: the most common alterations found in bacterial, viral, and fungi gut populations in IBD patients. IBD: Inflammatory bowel disease; *E. coli*: *Escherichia coli*.

**Figure 2 microorganisms-12-01755-f002:**
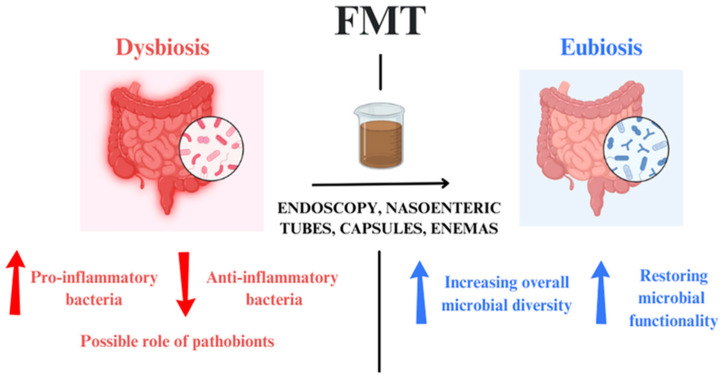
How FMT treats dysbiosis: dysbiosis is characterized by an increase in pro-inflammatory bacteria and a decrease in anti-inflammatory ones. On the other hand, eubiosis after FTM is guaranteed by an increase in microbial diversity. FMT: Fecal Microbiota Transplantation.

**Table 1 microorganisms-12-01755-t001:** FTM delivery methods: advantages and disadvantages.

FMT Delivery Methods	Advantages	Disadvantages
Enema	Inexpensive, well-tolerated, no sedation required, little procedural risk, can be performed by a non-physician	Not possible in poor rectal sphincter tone patients. Lowest efficacy rates in CDI [67]
Rectosigmoidoscopy	Less invasive, indicated for frail patients or patients at high risk of intestinal perforation	Expensive
Colonoscopy	Enables assessment of disease activity	Expensive, requires technical expertise
Naso-enteric tube	Inexpensive, no sedation required	Uncomfortable, less appealing for patients, requires radiological confirmation of tube placement, risk of vomiting, risk of aspiration
Upper endoscopy	Indicated in patients with severe colitis, patients who had lower GI surgery or without an intact colon	Expensive, sedation required, procedural risks
Capsule	More appealing to patients	Difficulty in production, necessity for delayed-release formulations

FMT: Fecal Microbiota Transplantation; CDI: Clostridioides difficile infection; GI: gastrointestinal.

**Table 2 microorganisms-12-01755-t002:** Selected evidence on FMT in UC patients.

Authors (Year)	Study Design		Sample Size	Treatment	FMT Characteristics	Concomitant Therapies	Main Outcomes	Safety Outcomes
Costello et al. (2017) [70]	Systemic review with meta-analysis	14 cohort studies	168 UC patients (from mild to severe UC forms)	FMT vs. placebo	Fresh donor stool (7 studies). Frozen donor stool (4 studies). NR (3 studies). Administrations’ numbers ranged from 1 to 6: via the upper GI tract (13%), via the lower GI tract (74%), and via both routes in 21 (12%).	Maintenance of regular IBD medications (except for 1 study that allowed only 5-aminosalicylates)	Clinical response: 93/168 (55%; 95% CI: 36.7–71.7%). Clinical remission: 39/168 (24%; 95% CI: 11–40%). Endoscopic remission (only assessed in 50% of the studies): 16/56 patients (29%).	The most common AEs were self-limiting GI complaints. No significant differences between the number/type of AEs between donor and placebo groups.
		4 RCTs	277 UC patients	FMT vs. placebo	Frozen donor stool (2 studies). Fresh donor stool (1 study). Fresh and frozen stool (1 study). The donor stool was processed anaerobically (1 study) or aerobically (3 studies).	Thiopurines, corticosteroids, 5-aminosalicylates, and biologic therapies	Clinical remission: 28% in the FMT groups vs. 9% in the placebo groups (OR: 3.67, 95% CI: 1.82–7.39, *p* < 0.01). Clinical response: 49% in FMT patients vs. 28% in placebo patients (OR: 2.48, 95% CI: 1.18–5.21, *p* = 0.02) [70].	
Paramsothy et al. (2017) [76]	Systemic review with meta-analysis	24 cohort studies	307 UC patients	FMT	Strong variation in FMT infusion methodology/protocol: different routes of administration, number and frequency of infusions, the dosage of stool per infusion, preparation of inoculum [fresh or frozen], and FMT donor source [related or unrelated].	Steroids, mesalamine, thiopurines, and biologic therapies	Pooled proportion of patients achieving clinical remission was 33% [95% CI = 23–43%].	No report of major AEs or SAEs that were deemed clinically related to FMT. Most reported AEs were transient minor GI complaints and/or fever.
		20 cohort studies	234 UC patients				Pooled proportion of patients achieving clinical response was 52% [95% CI = 40–64%].	
		4 RCTs	140 UC patients		Frozen donor stool (2 studies); fresh donor stool (1 study); fresh or frozen or combined (1 study)		Clinical remission: 36% [P-OR = 2.89, 95% CI = 1.36–6.13, *p* = 0.006]. Clinical response: 52% [P-OR = 2.48, 95% CI = 1.18–5.21, *p* = 0.016]	

UC: Ulcerative Colitis; CD: Crohn’s Disease; IBD: Inflammatory Bowel disease; CP: Chronic Pouchitis OR: Odds Ratio; AE: Adverse Events; SAE: Serious Adverse Events; CI: Confidence Interval.

**Table 3 microorganisms-12-01755-t003:** Selected evidence on FMT in CD patients.

Authors (Year)	Study Design		Sample Size	Treatment	FMT Characteristics	Concomitant Therapies	Main Outcomes	Safety Outcomes
Caldeira et al. (2020) [80]	Systemic review with meta-analysis	9 RCTs	UC, CD, and CP patients	FTM vs. placebo	Fresh donor stool (21 studies). Frozen donor stool (6 studies). Fresh or frozen (3 studies). Capsules (2 studies). NR (2 studies).	NR	Clinical remission: RR 1.70 [95% CI 1.12, 2.56]. Clinical response: RR 1.68 [95% CI 1.04, 2.72].	
		24 interventional studies					Overall clinical remission of 37%. Overall clinical response of 54%.	AE prevalence of 29%
		Proportions meta-analyses					CD remission rate of 47.6%; UC Remission rate of 35.0%; CP remission rate of 4%. No statistically significant difference was observed for the IBD-type subgroup analysis (*p* = 0.152).	CD AE prevalence: 5.8%; UC AE prevalence: 36.9%; CP AE prevalence: 29.9% (for any AE).
Cheng et al. (2021) [84]	Systemic review with meta-analysis (involving 1 RCT, 7 cohort studies, and 4 case studies)		106 CD patients	FMT	Fresh donor stool (7 studies); Frozen donor stool (4 studies); NR (1 study); Endoscopically administered (upper or lower GI tract).	NR	Pooled rate of clinical remission was 0.62 (95% CI 0.48, 0.81) (based on 11 studies). Pooled proportion of CD patients that achieved clinical response was 0.79 (95% CI 0.71, 0.89) (based on 7 studies) with low heterogeneity (I2 = 43%).	Most AEs were minor and self-resolving. No major FMT-related AEs were reported.
Tan et al. (2022) [85]	Systemic review with meta-analysis		IBD patients (438 UC and 17 CD)	FTM vs. placebo	Frozen donor stool (202 pt); Fresh donor stool (105 pt); Frozen or fresh stools (136 pt); frozen stools and oral capsules (12 pt); 395 pt via the lower GI tract, 48 pt via the upper GI tract.	NR	Clinical remission (based on 11 studies): RR: 1.44, 95% CI: 1.03 to 2.02, I^2^ = 38%, *p* = 0.03 with no significant heterogeneity. Clinical response (reported by 8 studies): RR 1.34, 95% CI: 0.92 to 1.94, I^2^ = 51%, *p* = 0.12, with moderate heterogeneity.	Most AEs were mild and self-resolving.

UC: Ulcerative Colitis; CD: Crohn’s Disease; IBD: Inflammatory Bowel disease; CP: Chronic Pouchitis OR: Odds Ratio; AE: Adverse Events; SAE: Serious Adverse Events; CI: Confidence Interval; pt: patients; NR: Not Reported.

**Table 4 microorganisms-12-01755-t004:** Selected evidence on FMT in pediatric IBD patients.

Authors (Year)	Study Design	Sample Size	Treatment	FMT Characteristics	Concomitant Therapies	Results	Safety Outcomes
Hsu et al. (2023) [88]	Systematic review with meta-analysis	352 pediatric IBD patients	FMT	From commercial stool banks, screened and unrelated donors, related donors	NR	1 month after FMT: the clinical response was achieved in 58.8% of patients; clinical remission was achieved in 64.7% of patients; 44.1% of patients achieved both	Pooled rate of AE: 29% (95% [CI]: 15.0%, 44.0%; *p* < 0.001; I2 = 89.0%, Q = 94.53). Pooled rate of SAEs: 10% (95% [CI]: 6.0%, 14.0%; *p* = 0.28; I2 = 18.0%, Q = 9.79)
Fang et al. (2018) [83]	Systematic review with meta-analysis	67 pediatric IBD patients (47 UC and 20 CD) and 252 adult IBD patients (178 UC and 74 CD)	FMT	Fresh donor stool, frozen donor stool, or NR	Steroids, immunomodulators, biologic therapy	Pooled estimate of clinical remission of FMT was: 10% for pediatric UC (95% CI: 0–43%), 45% for pediatric CD (95% CI: 24–66%), 26% (95% CI:10–48%) for adult UC; 22% (95% CI = 3–52%) for adult CD	FMT was tolerable and safe for IBD. Reported AEs: fever, abdominal pain, bloating, diarrhea, nasal congestion, vomiting, and sore throat (most of which are self-limiting, lasting no more than 24 h)

UC: Ulcerative Colitis; CD: Crohn’s Disease; IBD: Inflammatory Bowel disease; OR: Odds Ratio; AE: Adverse Events; SAE: Serious Adverse Events; CI: Confidence Interval; NR: Not Reported.

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
