# Peer review of "The Role of Fecal Microbiota Transplantation in IBD"

_microorganisms, 2024, doi:10.3390/microorganisms12091755_

Round 1

Reviewer 1 Report

Comments and Suggestions for Authors

In the attachment

Comments on the Quality of English Language

Minor editing of English language required.

Reviewer 2 Report

Comments and Suggestions for Authors

The review paper by Fabrizio Fanizzi is informative and interesting. I have the following questions and comments:

1, the figure legend of Figure 1 and Figure 2 are the same. Please revise. 

2, line 115, should be "either as a stand-alone therapy ".

3, why FMT in IBD has not matched the clear efficacy observed in treating CDI? The authors should further discuss the underlying mechanisms. In my opinion, gut dysbiosis is only one contributor to the development of IBD, but for CDI, it is almost the only contributor.

Round 2

Reviewer 2 Report

Comments and Suggestions for Authors

The authors have revised the manuscript accordingly. It can be considered for publication.